# Effects of Channel Length Scaling on the Electrical Characteristics of Multilayer MoS_2_ Field Effect Transistor

**DOI:** 10.3390/mi14020275

**Published:** 2023-01-20

**Authors:** Sreevatsan Radhakrishnan, Suggula Naga Sai Vishnu, Syed Ishtiyaq Ahmed, Rajagopalan Thiruvengadathan

**Affiliations:** 1SIERS Research Laboratory, Department of Electronics and Communication Engineering, Amrita School of Engineering, Amrita Vishwa Vidyapeetham, Coimbatore 641112, India; 2Mechanical Engineering Program, Department of Engineering and Technology, Southern Utah University, Cedar City, UT 84720, USA

**Keywords:** MOSFET, MoS_2_, 2-D material, transition metal dichalcogenides, quantum transport, COMSOL, electron transport, MATLAB, computational

## Abstract

With the rapid miniaturization of integrated chips in recent decades, aggressive geometric scaling of transistor dimensions to nanometric scales has become imperative. Recent works have reported the usefulness of 2D transition metal dichalcogenides (TMDs) like MoS_2_ in MOSFET fabrication due to their enhanced active surface area, thin body, and non-zero bandgap. However, a systematic study on the effects of geometric scaling down to sub-10-nm nodes on the performance of MoS_2_ MOSFETs is lacking. Here, the authors present an extensive study on the performance of MoS_2_ FETs when geometrically scaled down to the sub-10 nm range. Transport properties are modelled using drift-diffusion equations in the classical regime and self-consistent Schrödinger-Poisson solution using NEGF formulation in the quantum regime. By employing the device modeling tool COMSOL for the classical regime, drain current vs. gate voltage (I_D_ vs. V_GS_) plots were simulated. On the other hand, NEGF formulation for quantum regions is performed using MATLAB, and transfer characteristics are obtained. The effects of scaling device dimensions, such as channel length and contact length, are evaluated based on transfer characteristics by computing performance metrics like drain-induced barrier lowering (DIBL), on-off currents, subthreshold swing, and threshold voltage.

## 1. Introduction

As the miniaturization of traditional silicon-based integrated chips has increased to improve performance and reliability, geometric scaling of critical dimensions is indispensable. However, in the forthcoming decade, the classic technological underpinnings are anticipated to cease as the lithographic requirements reach atomic levels, and thus, a practical limit on silicon miniaturization [1]. The ITRS roadmap, which recorded growth over the past four decades, has predicted no improvements beyond 2021 [2]. The major issues associated with traditional CMOS scaling resulted from severe mobility degradation in a short channel regime [3,4].

On another related development, intense research efforts undertaken since the 1990s have enabled the scalable synthesis of novel nanomaterials with precise control of their structural, thermal, mechanical, and electrical characteristics [5,6]. Among the wide variety of nanomaterials, carbon nanotubes (CNTs), graphene, and nanocomposites formed with inclusions of CNTs and graphene are very attractive from an electronic device perspective, as well for other applications, owing to the possibility of tuning their electronic and electrical characteristics [7,8]. From the viewpoint of electronic devices, 2D materials, including graphene and transition metal dichalcogenides (TMDs) such as MoS_2,_ offer unique advantages in terms of scaling and ultra-thin body. In comparison to graphene and other TMDs, MoS_2_ is considered a frontrunner due to higher mobility, better gate control, tailorable bandgap, thermal stability, and scalability to lower dimensions. Comprehension of the properties as a function of scaling the device parameters is indispensable as this allows for the compatibility of these materials with current CMOS technology. 

Recently published works have demonstrated the facile fabrication of nano-FETs by employing the advantages offered by MoS_2_. The major work on MoS_2_ as an alternate material was built by Radisavlejvic et al., who demonstrated high mobility monolayer (ML) with high on/off currents [9]. Han et al. address the design challenges and insights into the dielectric integration, Fermi level pinning of metal contacts, and scaling, further emphasizing the need for lower contact resistance at the metal-semiconductor junction and quality interface between dielectric and channel [10]. Han et al. also demonstrated that the metal-semiconductor junction in MoS_2_-Ni/Au interface is Schottky limited, and few selected metals like Ni and Au are more suited due to near-matching values of the bandgap-work function. Cao et al. presented a model of monolayer FETs analyzing doping effects, mobility degradation effects, and interface trap density (Dit) [11]. Works to integrate MoS_2_ models in an analog circuit are presented by Wei et al. using a ring oscillator set-up [12]. Zhang et al. showed a comparison between a MoS_2_FET vs. conventional Si-MOSFETs by comparing the performance of delay parameters and devised a parametric model for the same [13]. Pellagini et al. described modeling FETs using an atomistic solver in combination with COSMOL models in view of analyzing the effects of defects [14]. Aruthchelvam et al. performed a statistically significant sample of scaling models down to channel length of 30 nm [15]. Even though a plethora of work has been done on the scaling of transistors, research efforts have yet to be invested in MoS_2_-based MOSFETs, specifically to meet the current technological nodes of sub-10 nm. 

Concomitantly, work has been done on incorporating quantum mechanics in device modeling of sub-15 nm nodes for other materials. To investigate device characteristics of MoS_2_FETs, the device’s properties are simulated using the non-equilibrium Green’s function (NEGF) method [16,17,18,19], which solves Schrödinger’s wave equation under non-equilibrium conditions, assuming periodic boundary conditions. A variety of devices including Si MOSFETs [20], novel material MOSFETs [21], and CNTFETs [22] have been investigated using the NEGF method. 

In this work, the authors present a detailed study of scaling 4-layer (4 ML) MoS_2_-FETs down to the sub-10-nm channel length regime and analyzing the short channel effects that result thereby. For the simulations and analysis of FETs with channel lengths ranging from 34 nm down to 15 nm, the authors have employed a classical drift-diffusion model; whereas for channel lengths less than 15 nm down to 8 nm, due to ballistic conduction in this regime [16,17,18,19,20,21,23], the authors have invoked the concepts of quantum mechanics to study and analyze scaling, as will be discussed in the work. The simulation and analysis of geometric scaling down to the sub-10 nm range are performed by employing the device-level simulation software COMSOL in combination with MATLAB to incorporate quantum effects. In this work, the devices are evaluated in terms of performance metrics like the I_on_/I_off_ ratio, Subthreshold Swing (SS,) and threshold voltage (V_th_). Devices having a higher I_on_/I_off_ ratio are preferred as this would be a key factor in determining the switching characteristics of CMOS. Ideally I_off_ must be as low as possible to minimize wasted leakage power in the OFF state. Subthreshold Swing (SS) is also a key factor in evaluating these models as this determines the roll-off rate in the sub-threshold region. This is a measure of the rate at which the charge carrier diffuses from the channel region when the device turns off. The threshold voltage (V_th_) is another measure for performance evaluation that determines the voltage at which the transition from linear to saturation current occurs. In the next section, we present the modeling methodology employed to determine electron transport in classical (for channel length L_ch_ > 15 nm) and quantum (L_ch_ < 15 nm) regimes. The results and discussion section explains the simulated data interpreted in terms of the physics of voltage transfer characteristics (VTC), in addition to addressing the future scope.

## 2. Computational Modeling & Methodology

A schematic of the MoS_2_ FET structure, the materials used, and the physical parameters used in this simulation work are shown in Figure 1.

In this work, the authors chose the global back-gate biasing device configuration to have V_th_ tunability (modulating V_th_) for low-power applications. The following considerations were made in choosing the device materials of the model. Previous work done on metal-semiconductors junctions establishes that elements like titanium, platinum, and nickel form Schottky-based contacts [24]. Herein, a metal-semiconductor junction is modeled as Schottky, based on the work function difference between 4 ML MoS_2_(5.36 eV) and Ni (5.01 eV) contact. Electron affinity (χ^o^) of 4 ML MoS_2_ is taken to be 5 eV. The metalwork function and Richardson constant are given as input to the model [10,24]. The potential at the contact is defined as
(1)$$V=V_{0}+\phi_{B}-\;\chi^{o}-V_{eq}$$
where V_0_, *ϕ*_B_ and V_eq_ are applied potential, the work function of metal and offset in Fermi level in terms of potential applied respectively. An N-type dopant concentration of 10^18^ cm^−3^ is used in the simulation.

The simulations are performed in a 4-multilayer (4 ML) thick MoS_2_ with an average thickness of 2.7 nm. A 4 ML MoS_2_ is chosen considering the trade-off between bandgap and field-effect mobility. To elaborate, devices with smaller bandgap (>10 ML MoS_2_) result in a larger Schottky barrier height value at the contact (non-ohmic contact) but would show excellent channel mobility (>45 cm^2^/Vs). In contrast, devices that form ideal ohmic contact require a larger bandgap (1–2 layers of MoS_2_) but show very low values of mobility (<20 cm^2^/Vs) [25], as neither too large a bandgap nor very low mobility as the material of choice. The dielectric is chosen to be high-k HfO_2_ as this shows improved effective oxide thickness resulting in higher transconductance with minimal leakage. Replacing the polysilicon-based global back-gate with metal-like materials is important to avoid increased threshold voltage due to poor gate-dielectric bonding and non-uniform surface trap charges. In order to mitigate this issue, metal-like TiN is the material of choice for the back-gate, as this has a mid-gap work function along with its tunable work function with film thickness. The gate oxide used is 1 nm HfO_2_ which is placed over a thin TiN back-gate contact (~1 nm). Ni is chosen as the source and drain contact material, as the Ni-MoS_2_ interface shows minimal Schottky barrier height (SBH), thus having implications on contact resistance but without affecting mobility. All other input parameters to the model are listed in Table 1. The mobility value used in this model agrees with the value mentioned in a published work that too states mobility of 42 cm^2^/Vs for multilayer (few tens of nm thick devices> 15 layers) devices owing to the limitations in channel-contact interface The interfaces are assumed to be ideal, and hence edge effects as well as Dit (interface trap density) effects are not taken into account in the computational simulation method adopted here. The authors have assumed a 1-D model as the conduction is near-ballistic.

While sweeping L_ch_ (8 nm to 34 nm) and L_cont_ (2 nm to 200 nm), the lateral width (W) has been varied as 2 nm, 20 nm and 100 nm separately investigated and studied. The simulated data is then normalized with respect to lateral dimension for the chosen width. 

Models with wide channel width but of small L_ch_ are neglected as they are impractical and would show an extreme parasitic drop in voltage across the resistance due to contact. The threshold voltage (V_th_) has been traditionally determined by employing one of the three methods, namely, (i) linear extrapolation of peak-mobility (to avoid second-order effects) [27], (ii) linear extrapolation of peak-transconductance [28], and (iii) constant current measurement (V_th,CC_ extracted at I_D_ = 10* W/L_ch_ in units of nA) [15]. Here, the authors determine the threshold voltage by the constant current method following the work reported in the literature [29]. The subthreshold swing is taken to be the minimum value across the entire range. Drain-induced barrier lowering (DIBL) is evaluated using V_th_ values evaluated at V_D,low_ = 0.05 V and V_D,high_ = 1 V. The I_off_ current corresponds to the current at a fixed displacement field (vertical field) of 0.4 V/nm below V_th,CC_ (i.e., |V_GS_ − V_th_,_CC_|/EOT = 0.4 V/nm), where V_th_,_CC_ denotes the threshold voltage under constant current condition and EOT denotes the equivalent oxide thickness, while the I_on_ current is taken to be saturation current [15]. 

As mentioned earlier, the authors have employed COMSOL Multiphysics to study the transfer characteristics for devices with a channel length ≥15 nm. On the other hand, previous work reveals that the near ballistic transport due to tunnelling plays a vital role in devices with channel length <15 nm [30,31,32]. Therefore, the quantum transport model is invoked and is modelled using MATLAB. The equations in the above models are explained briefly. 

### 2.1. Drift-Diffusion Model

The authors employ the full-wave solver COMSOL Multiphysics, which provides a semiconductor module that contains various functionalities, enabling facile modeling and simulation studies of 2D-FETs. Even though finite element-based works were previously performed to include quantum-based effects [23], the authors opt for finite-volume method computation solvers because this approach provides a more accurate approximation of the fields in the element, discretization, and constraint on the local element and is thus more suitable for a potential distribution-based system. The steps involved in modeling the device using COMSOL are summarized in Figure 2.

Efficient physics-based meshing is customized to be well-refined at the interface. This discretization is then followed by assigning solver configuration, including termination condition, error floor, and choosing the solver method for convergence. Then guiding equations are fed as input by choosing the suitable elements. Equations used by the benchmark module are provided. 

This module computes using Poisson’s equation, which relates charge density (ρ) and potential (*V*), given by:(2)$$\nabla \cdot \left(-\varepsilon_{o}\varepsilon_{r}\nabla V\right)=\rho$$
where ε_o_ is the permittivity of free space and ε_r_ is the relative permittivity of the medium, respectively.

Complete ionization of the module is assumed and based on a semi-classical model; charge is computed by the following the relation
(3)$$\rho^{+}=q\left(p-n+N_{d}^{+}-N_{a}^{-}\right)$$
where *q*, *p*, *n*, $N_{d}^{+}\;$ and $N_{a}^{-}\;$ are the charge of an electron, the carrier concentration of electrons, the carrier concentration of holes, and the donor and the acceptor impurity concentrations, respectively.

The semi-classical model is used to describe the metal-semiconductor interface. The charge densities of electrons and holes should follow the continuity equation in accordance with Gauss’s law, which can be expressed in the form of the following equations
(4)$$\nabla \;\cdot \;J_{n}=0\;$$
(5)$$\nabla \;\cdot \;J_{p}=0$$

Here, J_n_ and J_p_ denote the electron and the hole current densities. The carrier current densities can be computed by employing the drift-diffusion model and are expressed as:(6)$$\;J_{n}\;=n{µ}_{n}\nabla E_{c}+{µ}_{n}k_{B}T\nabla n+qnD_{n,th}\nabla \;lnT$$
(7)$$J_{p}=p{µ}_{p}\nabla E_{v}-{µ}_{p}k_{B}T\nabla p-qpD_{p,th}\nabla \;lnT\;$$
where *µ_n_*, *µ_p_*, *D_p,th_*, *D_n,th_*, *T* and *k* are the mobility of electrons, mobility of holes, the thermal diffusion coefficient of holes, the thermal diffusion coefficient of electrons, temperature, and Boltzmann constant, respectively. As can be observed from the equations for carrier current densities, terms indicating contributions from drift, diffusion, and thermal effects are present [33].

### 2.2. Quantum Transport

The classical models of simulation are optimal for bulk devices but for mesoscopic and nanoscopic electronic devices, quantum mechanical models are required. For L_ch_ < 15 nm, the conduction in channels behaves at near ballistic limits, and the generation of the electron-hole pairs is governed by the band-to-band tunneling despite thermal effects and hence displays quantum effects [34]. The atomicity/molecularity of the device alters the quantum behavior, which affects the potential of the system. Therefore, to include quantum confinement effects and band structure contributions, we invoke quantum modeling of the device. To investigate the device characteristics of MoS_2_FETs, the properties of the device are simulated using the non-equilibrium Green’s function (NEGF) method, which solves Schrödinger’s wave equation under non-equilibrium conditions with periodic boundary conditions. Here, the Hamiltonian of such devices has been obtained by employing the Tight-Binding approach with overlapping integrals of neighboring atoms [35,36]. Using the results of the Schrödinger scheme, a self-consistent scheme is developed in which the potential and electron charge density are iterated until convergence. In order to speed up this computation, Newton-Raphson is used to check convergence and to update new values of charge density and self-consistent potential (U_sc_). The brief explanations of the Quantum model employed as well as algorithmic steps for a self-consistent scheme in the Schrödinger-Poisson design are detailed below.

Based on the following methodology, a simulation framework was developed in MATLAB, and the implementation follows the works of Datta et al. [16,17,18,19]. Consider a two-terminal device connected by a channel, with electric fields (e.g., gate) as well as charge flow (e.g., source/drain contacts). A self-consistent potential U_sc_ (which is computed by both Poisson’s and Schrödinger) is ‘self-consistent’ because it changes to adjust U_sc_ inside the device, which in turn changes charge density until consistent values are arrived at. For this iterative simulation using the Newton-Raphson method, two major equations are considered: the transport equation and the Poisson equation. The input of the Poisson equation is the charge density, and the output is the self-consistent potential which is computed using the Finite Difference method. A quantum model such as NEGF formalism is deployed in order to compute charge density in the set of equations. The self-consistent procedure for simulation comprises of the following steps [37]:Discretize all the operators using the Finite Difference method.For initial input of U_sc_, initiate a self-consistent loop (e.g., a potential U_sc_ of flat band condition). Boundary conditions at the edge nodes are mapped to gate voltage.Compute suitable Hamiltonian, contact self-energy matrices for the given source/drain terminal potential.The retarded Green’s function is computed to obtain the electron density matrix.The Poisson equation is solved for the self-consistent potential, U_sc_, using electron density computed in the previous step.Steps 3–5 are repeated until the convergence of potential is less than the specified tolerance.With the final electron density matrix and the corresponding self-consistent potential, the transport current of the device for the given gate voltage is calculated.Steps 2–7 are repeated for a sweep across all the V_GS_Execute steps 1–8 for V_DS_ = 0.05 V and 1 V.

For elaborate discussion on the Quantum model, interested readers are requested to look into Supporting info.

## 3. Results and Discussion

The steps taken in data extraction following computational simulation are summarized in Figure 3.

The transfer characteristics (VTC) for various channel lengths at a low drain voltage of 0.05 V (to avoid drain-induced effects) are presented in Figure 4. 

By sweeping the gate voltage at V_D_ = 0.05 V, the following observation can be made: The off-current increases when channel length decreases, while the on-current saturates independent of channel length. Accordingly, we extract the off-current values across various channel lengths, which are shown in Figure 5. 

This is hypothesized to be a consequence of poor gate control when the channel length is scaled down from 34 nm to 8 nm leading to increased leakage in the sub-threshold region.

From Figure 5, we observe that the scaling of the off-current is disproportional to the channel length, which suggests that the off-current at lower drain voltage is contact limited. This is confirmed to be Schottky limited from the band profile of the transistor. Since the interface between Ni-contact and multilayer MoS_2_ has a barrier height of 0.45 eV in the junction, there is an abrupt change at the interface, confirming Schottky-type contact. The non-linear dependence of the drain current at a lower V_D_ of 0.05 V suggests that the current is limited by the Schottky barrier at contacts for various gate voltages despite the accumulation of electrons in the channel (biased ON state). This confirms that the off-current is indeed contact-limited and that the increased leakage is the result of poor gate control.

Subthreshold swing (SS) is obtained from the transfer characteristics. It is noticed that the degradation of SS with a decrease in channel length has a strong correlation with a poor off-current value and the variation of SS is shown in Figure 6 at two drain voltages at V_D,low_ = 0.05 V and V_D,high_ = 1 V. For V_D,low_, we observe that values of SS are 59 mV/dec and 95 mV/dec at channel lengths of 34 nm and 8 nm, respectively. The reason for the higher sub-threshold swing at a lower channel length is again due to the presence of Schottky-limited contacts in combination with the increased effect of drain voltage. As with traditional MOSFETs, when scaled down, the influence of the induced lateral field increases as the flat-band region narrows down further. 

This DIBL-like effect worsens the SS of the device resulting in a subthreshold roll-off and poor gate control. However, the dependence of SS with respect to channel length is disproportional, unlike DIBL. The presence of Schottky limited contact further increases the SS at a lower channel length. The SS is determined by the change in the conduction band edge with respect to gate bias. This effect of Schottky limited SS is due to increased depletion region at the contacts, thus further inhibiting the thermionic transport of carriers from metal to channel, limiting the conduction resulting in the poor subthreshold swing. 

On-current is a key parameter to be considered for a FET as this determines the intrinsic delay of the device. Higher I_on_ allows for more drive strength and larger fan-out. We expect that device operation at the linear region will have an inverse channel length dependence for lower V_D_. Normally, the inverse dependence of I_on_ on channel length is expected, but such dependence is not seen when the channel lengths are scaled down below 15 nm as evident from Figure 7. 

This is because the dependence of on-current on the channel length for short channels is not limited by channel length but by the velocity saturation and tunneling across the narrow Schottky peak. We note that this could be the result of the source degrading the effect of gate control, resulting in band edge sharpening and thus forming a narrow flat-band region for shorter channel lengths [38]. At higher drain voltages, the effect of the lateral field increases. And when the conduction band is lowered further, the saturation current is determined by charge tunneling through the barrier rather than the carrier conduction, known as drain-induced band narrowing [23,39]. This tunneling current is independent of the channel length and is affected solely by the velocity saturation and intrinsic mobility in the channel.

Values of threshold voltage are obtained from VTC (Figure 4) and are shown in Figure 8. As with MOSFETs, we find that at lower values of drain voltage, there is an expected V_th_ roll-off due to the short channel effects that require only a smaller gate voltage to invert the channel because the channel is already partially depleted. In contrast to the conventional threshold plot at various channel lengths and drain voltages, V_th_ at V_D,high_ is lower than that at V_D,low_ when channel lengths are reduced below 10 nm. This can also be inferred from the DIBL plot shown in Figure 9, where we have the transition of DIBL into negative values in agreement with our trends in V_th_ values (Figure 8). The reason for the negative DIBL and improved V_th_ at higher V_D_ is due to the negative capacitance effect. 

This effect originated from the coupling of drain-effective gate voltage via gate-drain capacitance. This coupling is caused due to the ferroelectric nature of the oxide capacitance [32]. For small values of V_G_, as the drain voltage increases, the coupling effect increases, and this results in conduction dominated by a lateral field (formed due to drain-channel coupling). Therefore, the surface potential at the channel is negative when V_G_ is zero or negligibly small, resulting in a negative capacitance effect. Since the current in the sub-threshold is an exponential function of surface potential, there is a lower subthreshold current for V_D,high_ in comparison to that at V_D,low_, leading to a negative DIBL. This results in improved V_th_ at higher V_D_. In order to confirm the negative DIBL, the simulation was repeated with a thicker dielectric of 4 nm HfO_2_. 

The results confirm the presence of negative DIBL which is shown in Figure 9. These observations agree well with the inferences reported in previous publications [30,32]. 

Scaling of contact length for a given channel length as in Figure 10 shows no deviation with respect to the on-current, suggesting that the scaling of contact length does not have any effects on other electrostatic properties. This sweep across contact length for all channel lengths, drain voltages, and widths confirms this observation is generic. This also implies that the underlying Schottky contact resistance would be independent of contact length, and scaling contact length would not alter the electrostatic parameters. Therefore, the proportional scaling of contact length would not degrade the performance of the transistor.

## 4. Conclusions

In this work, scaling effects of channel length (from 35 nm to 8 nm) on transistor performance are analyzed from I-V characteristics, which, in turn, are simulated employing a classical drift-diffusion model for channel lengths above 15 nm and quantum mechanics via self-consistent NEGF method below 15 nm taking inspiration from published works on Si-based MOSFETs. Due to the ballistic nature of carrier transport at a short channel regime, principles of quantum mechanics are used to understand the carrier transport behavior and thus the device parameters. The following conclusions are drawn based on the simulated data. First, the performance of the channel under consideration is established to be contact-limited, and the Schottky phenomenon at the contact determines the device behavior. The on-current of the system is determined primarily by drain-induced channel narrowing, source-drain tunneling, and saturation velocity; while the off-current has a strong dependence on coupling at high V_D_ (>~4 kT/e), resulting in negative capacitance effects. This results in higher V_th_ and negative DIBL at higher V_D_ in comparison to their lower V_D_. Lastly, we have established that the scaling of contact has no implications on any electrostatic parameters and thus scaling contact length would not affect transistor performance. 

The design challenges and expected trends are reasoned out in the work presented. Some key insights into the implementation of the design can be gained from the work. Scaling of contact length does not affect the performance of the system, but a contact resistance drop has a big implication on device performance. Therefore, design considerations for such small channel devices should be ensured to have the minimal parasitic drop across the contact resistance and remain an active area of research. Also, C-V curve-based extraction can be performed to arrive at a precise channel transport mechanism. Transfer line measurement can be performed to calculate interface trap density (Dit), which can further be used to verify the sub-threshold operation. Further, the C-V graphs can be used to understand field effect mobility, and this can be used to find the saturation velocity of the device. Apart from the design challenges, work can be done on exploring various doping profiles to improve gate control. Bandgap engineering can be performed to improve the mobility of the device. Alternatively, improved gate control possibilities using other gate oxides can be explored. Devising methods to overcome Schottky barrier limited channel can result in an improved subthreshold at lower channel lengths. Therefore, more work has to be done on engineering semiconductor-metal interface and improving the mobility in order to make the MoS_2_ based FETs an alternative to traditional Si-FETs.

Interested readers are requested to refer to Appendix A for an elaborate discussion on the Quantum model employed in this computational simulation work. 

## Figures and Tables

**Figure 1 micromachines-14-00275-f001:**
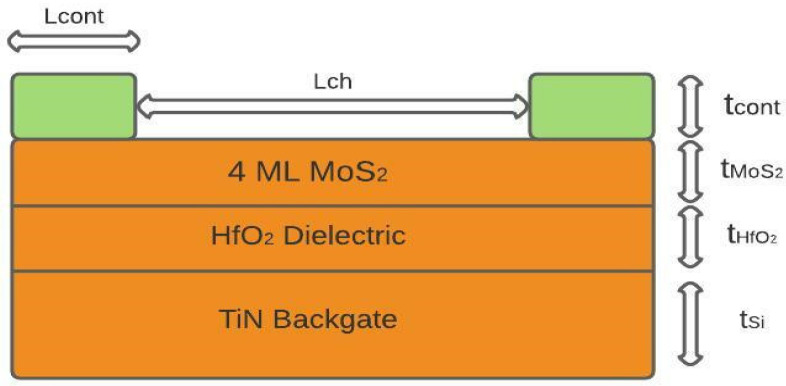
A schematic of the MoS_2_FET structure used in this simulation work.

**Figure 2 micromachines-14-00275-f002:**
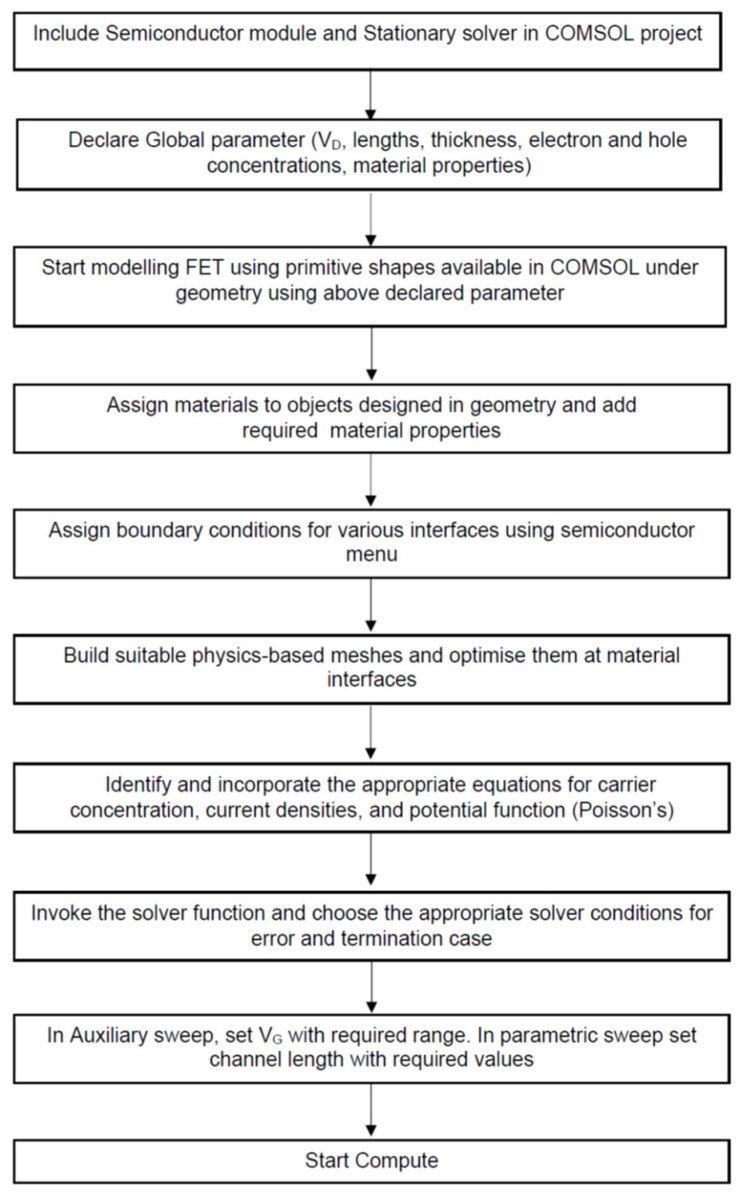
A schematic of the step-wise procedure followed to set up the computational simulation work.

**Figure 3 micromachines-14-00275-f003:**
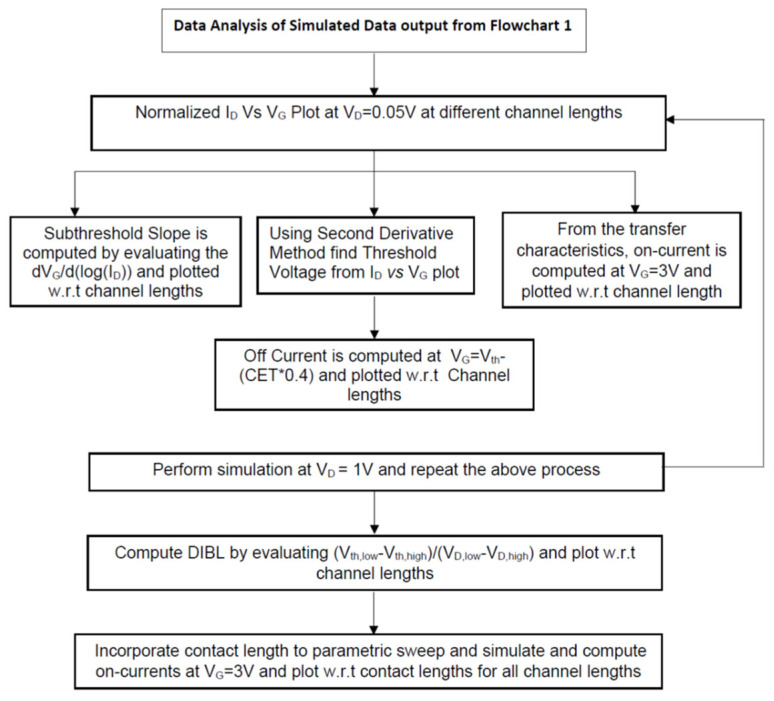
A flowchart depicting the steps followed to arrive at the derived plots using COMSOL.

**Figure 4 micromachines-14-00275-f004:**
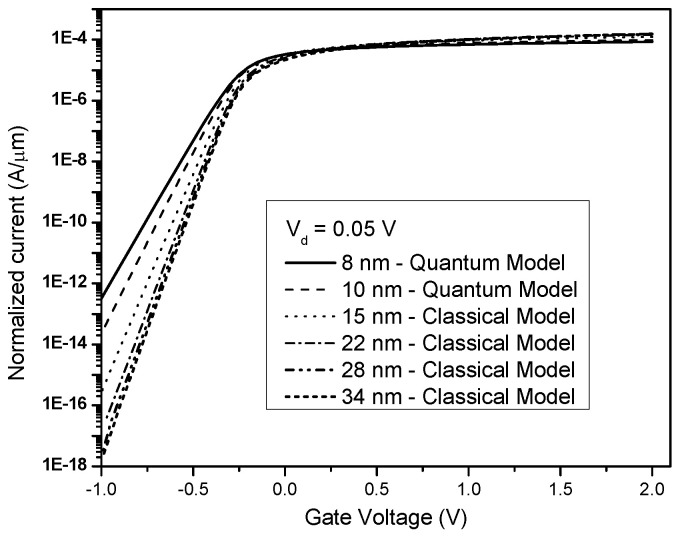
Transfer Characteristics at V_D_ = 0.05 V for various channel lengths.

**Figure 5 micromachines-14-00275-f005:**
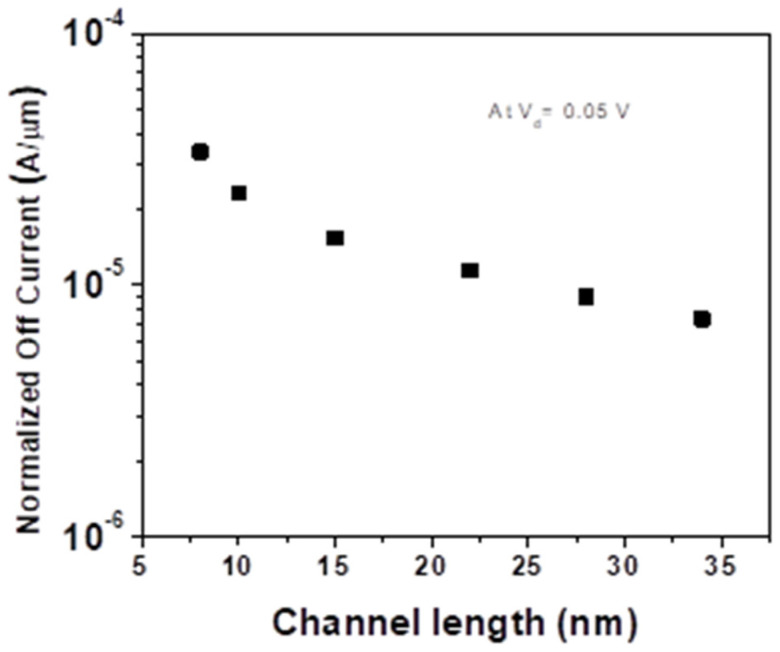
Normalized off-current (in A/µm) vs. channel length (in nm). This plot shows that off-current increases with a decrease in channel length as a result of poor gate control.

**Figure 6 micromachines-14-00275-f006:**
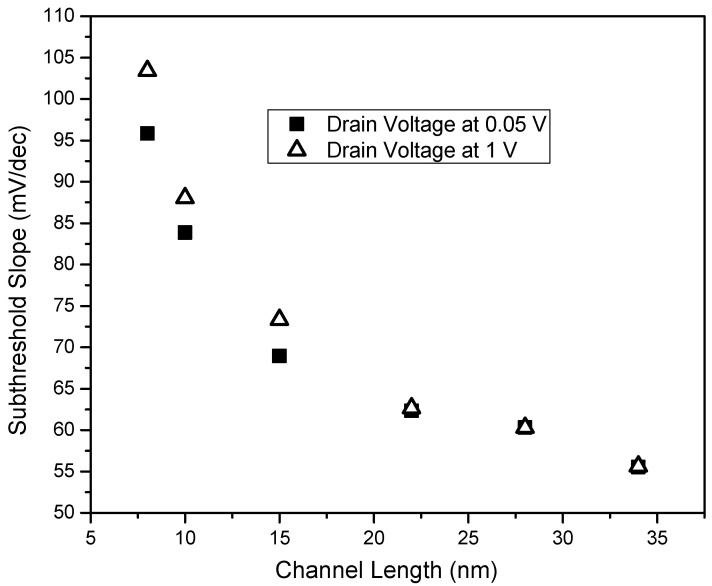
Subthreshold swing (in mV/dec) vs. channel length (in nm) at V_D_ = 0.05 V (in a square) and V_D_ = 1 V (in a triangle). This increase in SS as channel length decreases is the result of the increased contribution of drain voltage and increased depletion at the contacts.

**Figure 7 micromachines-14-00275-f007:**
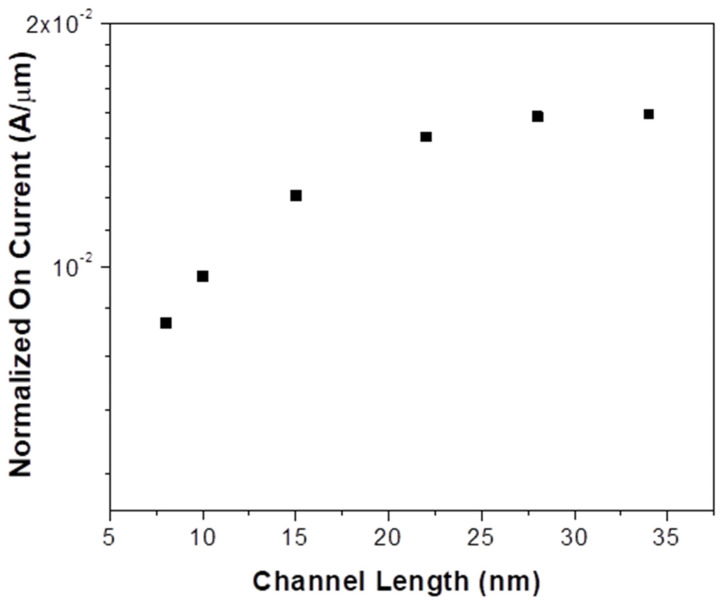
Normalized On-current (in A/µm) vs. channel length (in nm). This plot suggests that the on-current is independent of channel length and is determined by the tunneling path and velocity saturation.

**Figure 8 micromachines-14-00275-f008:**
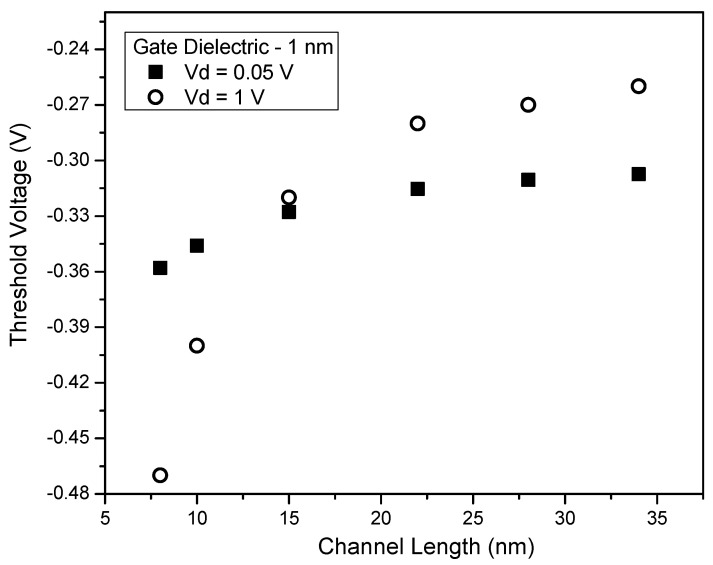
The threshold voltage (in V) vs. Channel length (in nm) at V_D_ = 0.05 V (in a square) and V_D_ = 1 V (in a circle). Note the transition point around 12 nm implying that below this channel length, the V_th_ of the device at V_D,high_ is lower than that at V_D,low_.

**Figure 9 micromachines-14-00275-f009:**
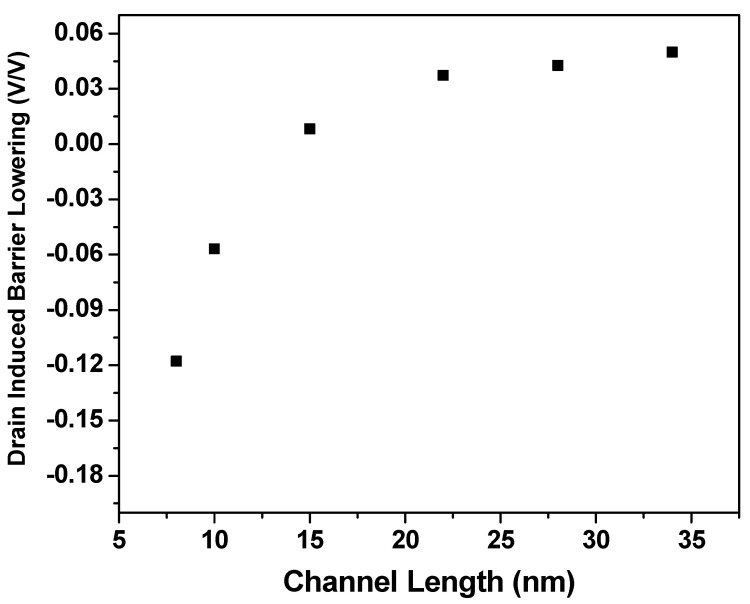
Effect of DIBL (in *v*/*v*) vs. channel length (in nm). This transition into negative values of DIBL for smaller channel lengths (<12 nm) is the result of drain-gate coupling.

**Figure 10 micromachines-14-00275-f010:**
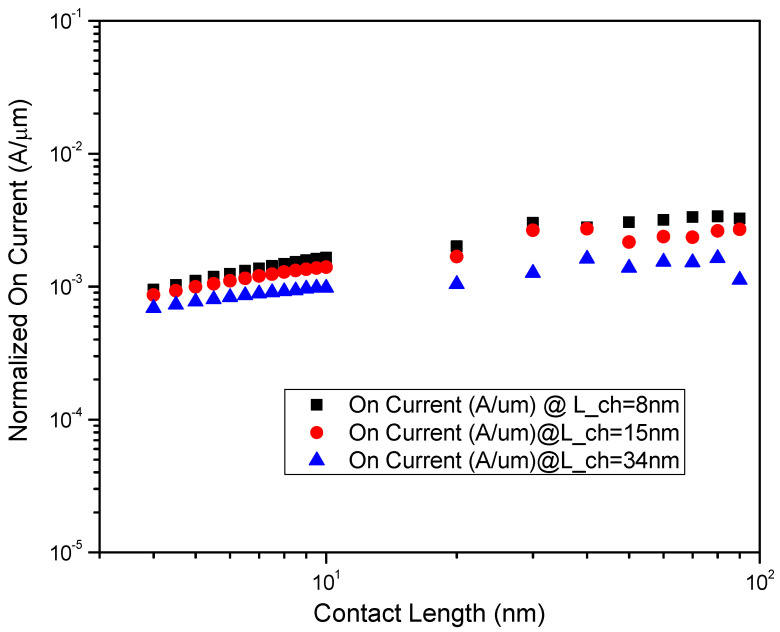
Normalized On-Current (in A/µm) vs. Contact length (in nm) for Lch = 10 nm. No dependence of Ion on Lcon suggest that carrier injection is only prevalent at the edge of the metal directly into the MoS2 channel.

**Table 1 micromachines-14-00275-t001:** Input parameters used in this simulation work. Data from [14,25,26].

Parameter	Value
Thickness of MoS_2_	0.65 nm/Layer
Bandgap 4 L MoS_2_	1.6 eV
Electron affinity	5 eV
Relative permittivity MoS_2_	11.3
Mobility	40 cm^2^/Vs
Drain and Source contact type	Schottky
Electron effective mass	0.56 m_o_
Hole effective mass	0.45 m_o_
HfO_2_ Thickness	1 nm
Work function of drain contacts (Ni)	5.01 V
HfO_2_ Relative permittivity	25
TiN back-gate thickness	~1 nm
SRH-time	0.3 ns
Donor concentration	10^18^ cm^−3^

## Data Availability

All of the simulated data and input parameters used in the computational simulation work have been made available within the manuscript.

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
