# Peer review of "Effects of Channel Length Scaling on the Electrical Characteristics of Multilayer MoS2 Field Effect Transistor"

_micromachines, 2023, doi:10.3390/mi14020275_

Round 1

Reviewer 1 Report

The comments have been attached as a pdf file.

Author Response

Please find the attached document that provides our elaborate response. The authors thank the reviewer for the valuable suggestions and comments. 

Reviewer 2 Report

The authors have presented work on 4ML MoS2 based FET Characteristics. I would recommend the work given that the authors clarifies a few major concerns. Kindly add a possible calibration of the proposed methodology with an experimental data. Is the considered simulation 1-D? How the authors considered the edge effect of the MoS2? Have the authors considered the effect of defects in MoS2 monolayer? There seems to be very low body effect, as the interface considered ideal? The OFF-current seems to be too high, kindly compare the proposed structure with the state-of-art devices.

Author Response

We thank the reviewer for the valuable suggestions and comments. 

We provide a summary of how we have addressesed the comments. Please see the attached document. 

Thank you. 

Round 2

Author Response

The authors thank once again the reviewer for the comments and suggestions in the first and the second round of review. The revisions as suggested by the reviewer certainly enabled to improve the scientific quality of the manuscript. 

Comment 1: I strongly recommended that the authors obtain the ID-VGS
characteristics and the band diagram along the device for L=10 nm and
25 nm at different drain voltage such as 0.05 and 1 V.

Response: The authors agree in principle with the reviewer for the inclusion of the suggested data. However, this can be done using advanced tool like TCAD Device Simulation tool. Unfortunately, we do not have access to this tool. This is primaily the reason for our computational approaches using COMSOL and MATLAB with NEGF. Besides we have cited all relevant and recent references reported in literature to support our findings in the manuscript. The authors expect that these simulation results will give some directions towards further research.

Comment 2: The English writing of the manuscript must be improved.
There are still many misspellings in the text; for example, the following
sentence shows such a misspelling. 

Response: The authors have performed proof reading of the manuscript with Grammarly software tool besides the manuscript has been proof-read by native English speaking person. This has helped us to correct the typo and also correct the other language errors such as verb, punctuation etc.

The authors thank once again for the comments and suggestions

Reviewer 2 Report

I would recommend the work as the authors have clarified most of the concerns.

Author Response

The authors thank once again the reviewer for the comments and suggestions in the first round of review. The revisions as suggested by the reviewer certainly enabled to improve the scientific quality of the manuscript. 

The authors have performed proof reading of the manuscript with Grammarly software tool besides the manuscript has been proof-read by native English speaking person. This has helped us to correct the typo and also correct the other language errors such as verb, punctuation etc.